# Evaluation of Social Cognition Measures for Japanese Patients with Schizophrenia Using an Expert Panel and Modified Delphi Method

**DOI:** 10.3390/jpm11040275

**Published:** 2021-04-06

**Authors:** Hiroki Okano, Ryotaro Kubota, Ryo Okubo, Naoki Hashimoto, Satoru Ikezawa, Atsuhito Toyomaki, Akane Miyazaki, Yohei Sasaki, Yuji Yamada, Takahiro Nemoto, Masafumi Mizuno

**Affiliations:** 1Department of Psychiatry, National Center of Neurology and Psychiatry Hospital, Tokyo 187-8551, Japan; hokano@ncnp.go.jp (H.O.); kubotar@ncnp.go.jp (R.K.); yujiyamada@ncnp.go.jp (Y.Y.); 2Translational Medical Center, National Center of Neurology and Psychiatry, Department of Clinical Epidemiology, Tokyo 187-8551, Japan; ysasaki@ncnp.go.jp; 3Department of Psychiatry, Hokkaido University Graduate School of Medicine, Sapporo 060-8638, Japan; hashinao@med.hokudai.ac.jp (N.H.); toyomaki@gmail.com (A.T.); a-miyazaki@med.hokudai.ac.jp (A.M.); 4Endowed Institute for Empowering Gifted Minds, University of Tokyo Graduate School of Arts and Sciences, Tokyo 153-0041, Japan; satoru-ikezawa@g.ecc.u-tokyo.ac.jp; 5Department of Neuropsychiatry, Toho University Faculty of Medicine, Tokyo 143-8541, Japan; takahiro.nemoto@med.toho-u.ac.jp (T.N.); mizuno@med.toho-u.ac.jp (M.M.)

**Keywords:** mental disorders, schizophrenia, developmental disorders, social cognition, social function, facial expression recognition, test battery, quality of life, systematic review, needs survey

## Abstract

Social cognition is strongly linked to social functioning outcomes, making it a promising treatment target. Because social cognition measures tend to be sensitive to linguistic and cultural differences, existing measures should be evaluated based on their relevance for Japanese populations. We aimed to establish an expert consensus on the use of social cognition measures in Japanese populations to provide grounds for clinical use and future treatment development. We assembled a panel of experts in the fields of schizophrenia, social psychology, social neuroscience, and developmental disorders. The panel engaged in a modified Delphi process to (1) affirm expert consensus on the definition of social cognition and its constituent domains, (2) determine criteria to evaluate measures, and (3) identify measures appropriate for Japanese patients with a view toward future quantitative research. Through two online voting rounds and two online video conferences, the panel agreed upon a definition and four-domain framework for social cognition consistent with recent literature. Evaluation criteria for measures included feasibility and tolerability, reliability, clinical effectiveness, validity, and international comparability. The panel finally identified nine promising measures, including one task originally developed in Japan. In conclusion, we established an expert consensus on key discussion points in social cognition and arrived at an expert-selected set of measures. We hope that this work facilitates the use of these measures in Japanese clinical scenarios. We plan to further examine these measures in a psychometric evaluation study.

## 1. Introduction

Schizophrenia is a severe mental disorder, and many patients with schizophrenia experience some degree of long-lasting functional impairment. One area that is impaired throughout the course of the disease is social cognition, which is defined as “the mental operations that underlie social interactions, including perceiving, interpreting, and generating responses to the intentions, dispositions, and behaviors of others” [1]. Social cognitive impairments directly affect patients’ social participation and capacity to build and maintain social relationships, thereby profoundly decreasing quality of life. This area has garnered considerable interest in recent years because the social cognition construct is believed to be more strongly linked to social functioning outcomes than traditional neurocognition [2], making it a promising treatment target.

With the emergence of social cognition as a major focus of schizophrenia research, numerous measures have been developed to assess its various aspects. However, the complexity and breadth of the social cognition construct, together with a lack of consensus regarding its constituent subdomains, has resulted in an overwhelming variety of measures based on differing theories and interpretations. Paradoxically, there is a dearth of widely accepted and standardized measures available for practical use. The Social Cognition Psychometric Evaluation (SCOPE) study aimed to establish a consensus on the theoretical structure of social cognition in schizophrenia and to systematically evaluate the psychometric properties of existing measures [3,4,5]. Four core theoretical domains of social cognition were established through expert surveys and RAND expert panel discussions: emotion processing, attributional style/bias, social perception, and theory of mind (ToM) [3]. Experts further identified the existing measures best suited to assess these domains. Two large-scale studies of schizophrenia patients and healthy control groups were subsequently conducted to examine the psychometric properties of 11 measures. Three measures showing particularly strong psychometric properties and associations with functional outcomes were recommended for use in clinical trials: the Hinting task [6], the Bell Lysaker Emotion Recognition Task (BLERT) [7], and the Penn Emotion Recognition Task (ER-40) [8].

The SCOPE study represents a significant step forward by providing a provisional battery of measures and a springboard for future endeavors. However, these results were based on data collected exclusively in the United States and may not be generalizable to different cultural contexts. Social cognition tasks are more sensitive to cultural and linguistic differences than neurocognitive tasks [9]. Stimuli for social cognition tasks often require the participant to understand social interactions. The “correct” interpretation of a social situation may be less obvious or even entirely different for people from a different culture. Stimuli may also include words or ambiguous dialogue with meanings that are not fully replicable across languages. Furthermore, there are believed to be baseline cultural differences in social cognitive ability and tendencies [10]. In short, the same measures established in the United States may not be suitable for assessing social cognition in other, particularly non-English-speaking, cultures. Thus, the cultural relevance and translatability of tasks must be individually considered for each culture [11].

Until recently, social cognition research in Asian populations has been limited to specific domains or been inconsistent in its choice of measures [12,13,14]. Following the SCOPE study, Lim et al. conducted a psychometric evaluation study examining a similar array of social cognition tasks with Singaporean schizophrenia patients and healthy controls [15]. All participants were fluent in English, and tasks were registered verbatim, without any modifications to the original English versions. The results were consistent with those of the SCOPE study in that the BLERT and ER-40 showed the strongest psychometric properties. However, contrary to the SCOPE study, the Hinting task showed less favorable characteristics. A possible explanation offered by the authors was that some of the vignettes used in the task could be culturally sensitive. These results suggest that, even with a shared language, social cognition tasks may show differing psychometric properties among populations with different cultural backgrounds. However, this study did not examine associations with neurocognitive and social functioning measures.

To our knowledge, no comprehensive psychometric evaluation studies in non-English-speaking populations have been conducted using either translated or originally non-English tasks. Such an attempt would face several new challenges. First, many social cognition tasks include ambiguous phrases or dialogue, which may be difficult to translate fully. Another factor is the anticipated correlation between familiarity with a culture and fluency in its language and its effect on task performance. In other words, in a typically non-English speaking country or cultural group, individuals fluent in English would be expected to have more insight into Western culture and thus may perform better on certain Western-developed tasks. The presence of such factors dictates the need to consider alternative social cognition measures than those originally developed in the Anglosphere.

The present study aimed to identify social cognition measures suitable for use in Japanese schizophrenia patients. It represents a pioneering attempt to systematically investigate the utility of social cognition measures for a non-English-speaking population. An expert panel was assembled and tasked with selecting a comprehensive group of measures that are relevant for the target population while also consistent with the abovementioned related studies.

## 2. Materials and Methods

### 2.1. Expert Panel Members

Expert panel members were recruited using a reputation-based snowball sampling procedure. Panel members were chosen from Japanese researchers performing psychological, neurobiological, psychophysiological, or neuroimaging research in the area of social cognition, broadly defined. Experts from fields other than schizophrenia were included to incorporate important concepts from closely related areas. Ultimately, nine experts in the areas of social psychology, social neuroscience, schizophrenia, and developmental disorders agreed to serve as panelists (Appendix A). No panelists reported financial conflicts of interest.

### 2.2. Key Discussion Points and Candidate Social Cognition Measures

We prepared a draft of items comprising key discussion points for establishing an expert consensus. This list included the definition and core domains of social cognition, the target population for the social cognition measures selected in this study, objectives for their use, and evaluation criteria for final recommendations following a psychometric evaluation study.

The definition of social cognition was quoted from the NIMH Workshop on Social Cognition in Schizophrenia [1], as this definition that had already garnered consensus from several experts in the SCOPE study [3]. We prepared a preliminary list of candidate social cognition measures to be considered by the expert panel. Measures were selected based on similar studies examining the psychometric properties of social cognition measures. The SCOPE study recommendations were given particular importance, although measures cited as promising but ultimately excluded were also reconsidered. In addition, the authors inspected the literature for relevant or promising social cognition measures that were originally developed in Japan or with pre-existing Japanese versions. The resulting list comprised 15 preliminary candidate measures, including all six measures recommended by the SCOPE study and two measures developed in Japan. The remaining measures were selected based on history of use in Japanese populations (Table 1). The principal investigators and secretariat then prepared a database listing the results of previous studies that examined the psychometric properties for each measure (Appendix A).

### 2.3. The Modified Delphi Process

This study used a modified Delphi process (RAND/UCLA appropriateness method) to (1) reaffirm consensus on the definition of the social cognition construct and its key domains, (2) establish criteria for evaluating the appropriateness of social cognition measures for use in Japanese populations, and (3) rate and select measures based on the established criteria with a view toward future psychometric evaluation studies (see Figure 1) [28,29]. This method was also chosen for the SCOPE study as a proven method to develop consensus-based test batteries, having been successfully used in the development of the MATRICS battery [30] and VALERO initiative [31] in the field of schizophrenia research [3]. We defined consensus as when the compilation of item statements reached approval of 80% or higher [32] in online voting sessions conducted via the *Google Forms* website. Panelists had approximately 2 weeks to complete each of the online surveys. Voting was repeated until consensus was reached on all items. After each round, iterative refinements were made to the item compilation based on participant feedback.

Panelists rated the appropriateness of each measure for use in Japanese schizophrenia patients based on the following criteria: (1) practicality of administration and tolerability for participants, (2) reliability, (3) utility, (4) convergent and criterion validity, and (5) international comparability. Panel members were provided with detailed descriptions of each measure, including psychometric data from the SCOPE study if available, along with a supplementary database of psychometric indicators for each measure that we compiled from the literature (Appendix A). Ratings were given on a 9-point scale, where 1 was “extremely inappropriate” and 9 was “extremely appropriate.” Panel members were also encouraged to provide feedback on individual items through a free form comment section. After each round, the results were compiled to prepare a summary document that presented the raw rating, mean, and median scores in histograms, together with individual comments gathered from each panel member. These documents were shared and used as a basis for the discussion rounds, where individual rating discrepancies were addressed. Certain points were agreed upon beforehand; 7 measures were to be selected from the 15 candidates for inclusion in a subsequent psychometric evaluation study, and the selected measures were to, as a whole, address as wide a range of social cognition domains as possible. Discussions were held in the form of online video conferences because of the COVID-19 pandemic and precautions regarding face-to-face group meetings and traveling.

## 3. Results

The final list of items agreed upon by the expert panel is shown in Appendix A.

### 3.1. Definition and Core Domains of Social Cognition

The panel agreed to maintain the well-known NIMH Workshop definition [1] and the four-domain structure of emotion processing, attribution style/bias, social perception, and ToM for social cognition in schizophrenia.

### 3.2. Target Population, Purpose of Use, and Evaluation Criteria of Social Cognition Measures

The target population for the social cognition measures selected in this study was Japanese schizophrenia patients. It was further specified in the panel discussions that the subsequent psychometric study would target “patients with schizophrenia whose symptoms have stabilized following the medication adjustment period in the acute phase and who are undergoing rehabilitation to improve social function.”

The initial focus of this study was to select measures that could be widely used in clinical practice. However, following discussion, the objectives were expanded to also consider the suitability of the measures for clinical trials.

A set of criteria to assess social cognition measures following the psychometric study was discussed and agreed upon among the panel. Feasibility and tolerability criteria were established in terms of administration time and participant ratings, respectively. Test–retest reliability would be considered acceptable with correlation coefficients greater than or equal to 0.6. Utility as a measure would be assessed in terms of floor and ceiling effects, with emphasis being placed particularly on the absence of floor effects because a task showing ceiling effects may still be useful for clinical purposes such as screening and aiding diagnosis. However, if a task is to be used as an outcome for interventional studies, the absence of both floor and ceiling effects across administration times was agreed to be favorable. Measures showing clear group differences between patients and healthy controls would be favored. Correlation with social function outcomes would also be emphasized. Incremental validity, or, in this case, increased predictive ability of social function outcomes beyond neurocognition, would also be given consideration. Finally, tasks recommended in the SCOPE study were agreed to be favorable in terms of international comparability. Grading criteria were modified so that grades would be considered for each purpose of use. Specific advantages and precautions for the use of each test would be described in the final article.

### 3.3. Panel Ratings and Selection of Social Cognition Measures

Descriptive statistics for the two rounds of panel ratings are provided in Table 2. A set of consensus measures was selected based on the final ratings. Seven tasks with the highest mean appropriateness ratings were selected: three tasks representing the emotion processing domain (the BLERT, ER-40, and Facial Emotion Selection Test (FEST)) and two tasks each for the domains of attributional style/bias (the Ambiguous Intentions and Hostility Questionnaire (AIHQ) and the Intentionality Bias Task (IBT)) and ToM (the Hinting Task and the Metaphor and Sarcasm Scenario Test (MSST)) (Table 3). No social perception tasks were included in the initially planned selection of seven tasks, prompting an additional discussion regarding whether the omission of a previously established core domain was acceptable. Ultimately, it was unanimously agreed to include two tasks representing social perception: The Social Attribution Task-Multiple Choice (SAT-MC) and the Biological Motion (BM) task. Thus, a total of nine measures representing each of the four established core domains comprised the final selection.

## 4. Discussion

Our primary aim was to identify social cognition measures appropriate for Japanese schizophrenia patients based on the opinions of experts in related fields, with a view toward future quantitative research. After establishing grounds for measure selection, the panel rated and discussed the suitability of 15 candidate measures (Table 2), ultimately arriving at nine measures representing all four domains (Table 3).

We first sought to obtain an expert consensus on the definition and theoretical framework of the social cognition construct. Our proposal of using the same four core domains established in the SCOPE study was met with some debate in the initial round of surveys. Several experts questioned the inclusion of the social perception domain, with concerns about the lack of clarity surrounding its definition and scope and an absence of well-established tasks. However, it was agreed that such shortcomings underscore the need for inclusion and further investigation of the construct. Other experts were concerned about the omission of metacognitive aspects. Nonetheless, the panel ultimately agreed to adopt the proposed four-domain structure, citing the importance of consistency and international comparability.

Initially, the target population was not specified to any stage of schizophrenia. However, it was pointed out that performance on social cognition tasks would vary significantly depending on what stage the patient was in and that such variance would make it difficult to adequately evaluate tasks’ psychometric properties. The panel agreed to narrow the target population to more stable patients, as they would also be the main targets for treatments to improve social functioning. The initial target population also included outpatients only. However, the panel discussed the need to address social cognitive dysfunctions in chronic patients hospitalized for reasons other than pure severity of symptoms, such as those in forensic psychiatric wards or patients with problematic behaviors not directly related to psychosis. Thus, the phrasing was modified to include such patients.

Two of the selected measures, the MSST and BM, are novel tasks that have yet to be systematically examined in the context of social cognition in schizophrenia. Furthermore, the MSST is a task that was developed in Japan. The panel unanimously agreed that the nature of this study as one of the first attempts to examine social cognition tasks in a non-English context dictates the need to consider tasks already established as suitable for Japanese populations. Although originally intended to evaluate autism spectrum disorder tendencies in children, it was agreed that the MSST could be applied to assess ToM in schizophrenia populations. The BM task has been mentioned in the literature in the context of incorporating social neuroscience paradigms into the field of social cognition [33,34] and specifically as a promising measure to explore the social perception domain [5]. The panel deemed the BM suitable for the particular objectives of this study due to its low dependence on cultural and linguistic factors, which suggests a high level of international comparability.

The AIHQ and SAT-MC were included despite being classified as “not recommended” in the SCOPE study [4,5]. The AIHQ comprises both open-ended, scorer-rated items and self-report Likert scales assessing participants’ responses to negative social situations. Answers for the open-ended questions are coded by two independent raters, which has been speculated to negatively affect psychometrics such as test–retest reliability. Buck et al. suggested that the psychometric properties of the AIHQ could be improved by expanding self-report items and removing the open-ended questions [35]. The Singaporean psychometric evaluation study conducted by Lim et al. showed more favorable results for the AIHQ, further suggesting its utility [15]. The SAT-MC was not included in the initially planned selection of seven tasks but was chosen as the highest rated among candidate social perception tasks. In the SCOPE study, the SAT-MC showed sub-par results in basic psychometrics such as test–retest reliability, owing largely to the use of two independent forms across the two administration times. The use of consistent test forms across administration times may produce more favorable results. In addition, given that the SCOPE study evaluated tasks based on suitability for clinical trials, the SAT-MC may potentially receive more favorable gradings when viewed through the lens of utility in clinical practice. Furthermore, the SAT-MC has long been considered less affected by linguistic and cultural differences than other social cognition tasks because it is non-verbal and less culturally loaded [26,36,37]. A recent cross-cultural study with South Korean and North American schizophrenia patients and healthy controls showed the SAT-MC to be consistent across groups and supported its utility across language and cultures [13], making it a strong candidate for our current study.

Notable omissions included the Eyes test and The Awareness of Social Inference Test (TASIT), which were both included in the SCOPE recommendations. The Eyes test was seen as possibly not suitable for Japanese populations due to cultural differences; in Japan, it is considered rude to stare at someone’s face and it is therefore not customary to read others’ emotions through their eyes. There was also further concern that the Eyes test significantly overlaps with emotion processing, despite being classified as a measure of ToM. The TASIT, which uses short but relatively complex video vignettes of actors enacting various social interactions, received lower ratings mainly due to concerns over translatability. Many experts also shared the opinion that certain task structures, such as the TASIT, are inherently more dependent on working memory, with performance on these tasks at risk of reflecting neurocognitive ability more strongly than social cognitive function.

This study is not without its limitations. First, it was largely influenced by the SCOPE study, and measures indicated in the SCOPE study therefore received more attention than others. We attempted to reduce these limitations by reconsidering measures not recommended by the SCOPE study and conducting a search of literature for Japanese-developed social cognition measures. Furthermore, our reliance on the SCOPE study for guidance meant inheriting its limitations regarding the social perception domain and lack of strong candidate measures to represent it. The expert rating results and ensuing discussions also suggested that the objectives of the present study may have inadvertently led to emotion processing tasks being favored because their simpler structures seemingly make them less vulnerable to changes in psychometrics caused by translation to Japanese. These observations were addressed in the panel discussions and ultimately influenced the decision to include a roughly equal number of measures from each of the four core domains. Another limitation is the relatively small number of experts recruited. Diversity regarding fields of expertise may have been somewhat limited, with relatively high weight on the field of schizophrenia and only one expert representing another clinical population (developmental disorders). Furthermore, this study may have benefited from including experts from other academic fields, such as cultural anthropology, to provide a more rigid examination of which tasks may or may not be appropriate for Japanese people.

The present study established an expert consensus on key discussion points and promising measures for assessing social cognition in Japanese schizophrenia patients. There is currently a lack of available information regarding the use of social cognition measures in a non-English-speaking cultural context. We hope that our research will inform and facilitate future endeavors in other countries. Subsequent phases of this study will involve a multi-center psychometric evaluation study in Japanese schizophrenia and healthy control populations using the expert-selected measures. However, a considerable portion of the selected tasks has yet to be validated in Japanese schizophrenia populations (BLERT, BM, and MSST) or even translated to Japanese (SAT-MC and IBT). A pilot study may be warranted to preliminarily confirm the utility and structural validity of the tasks and to identify any need for modifications. Cross-cultural studies comparing results among groups of different cultural and linguistic backgrounds may shed further light on which measures are more suitable for international comparison and collaborative research.

## Figures and Tables

**Figure 1 jpm-11-00275-f001:**
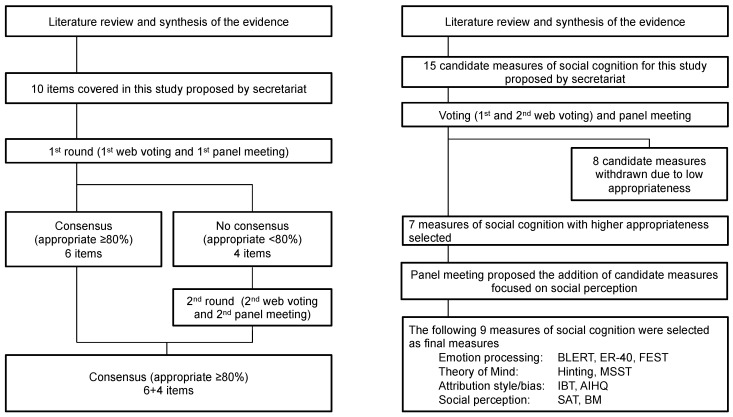
Overview of voting process.

**Table 1 jpm-11-00275-t001:** Candidate social cognition measures.

Domain/Measure	Original Citation	Total Citations (PubMed)	Citations Per Year (PubMed)
*Emotion Processing*			
Bell Lysaker Emotion Recognition Task (BLERT)	Bryson et al., 1997 [7]	44	1.91
Face Emotion Identification Test (FEIT)	Kerr and Neale, 1993 [16]	94	3.48
Noh Mask Test	Minoshita et al., 2005 [17]	2	0.13
Penn Emotion Recognition Test (ER-40)	Kohler et al., 2003 [8]	174	10.24
*Theory of Mind*			
Faux Pas Test	Stone et al., 1998 [18]	212	9.64
Hinting Task	Corcoran et al., 1995 [6]	145	5.8
Metaphor and Sarcasm Scenario Test (MSST)	Adachi et al., 2004 [19]	9	0.56
Reading the Mind in the Eyes Test (Eyes)	Baron-Cohen et al., 2001 [20]	864	45.47
The Awareness of Social Inference Test (TASIT)	McDonald et al., 2003 [21]	100	5.88
*Attributional Style/Bias*			
Ambiguous Intentions and Hostility Questionnaire (AIHQ)	Combs et al., 2007 [22]	59	4.54
Intentionality Bias Task (IBT)	Rosset, 2008 [23]	23	1.92
Social Cognition Screening Questionnaire (SCSQ) *	Roberts et al., 2011 [24]	(N/A)	(N/A)
*Social Perception*			
Biological Motion (BM) Task	Hashimoto et al., 2014 [25]	5	0.83
Social Attribution Task-Multiple Choice (SAT-MC)	Bell et al., 2010 [26]	20	2
Situational Feature Recognition Test (SFRT)	Corrigan and Green, 1993 [27]	(N/A)	(N/A)

***** Also measures Theory of Mind.

**Table 2 jpm-11-00275-t002:** Results of the expert panel ratings.

Domain/Measure	Median, Mean (SD)	
	1st Round	2nd Round
*Emotion Processing*		
ER-40	8, 7.1 (1.8)	8, 7.1 (1.7)
FEST *****	7, 7.2 (0.8)	8, 7.1 (1.9)
BLERT-J	7, 7.1 (1.6)	7, 7.3 (0.9)
Noh Mask Test	4, 4.2 (1.7)	3, 3.1 (0.9)
*Theory of Mind*		
MSST	8, 6.9 (1.9)	8, 6.8 (1.9)
Hinting	8, 7.2 (1.6)	7, 7.1 (1.6)
Eyes	5, 5.2 (2.4)	5, 5.0 (1.6)
Faux Pas	5, 5.0 (2.1)	5, 4.8 (1.8)
TASIT	5, 4.6 (2.8)	5, 4.3 (1.5)
*Attributional Style/Bias*		
AIHQ	7, 6.4 (1.8)	7, 6.2 (1.7)
IBT	6, 6.0 (1.9)	6, 5.8 (0.9)
SCSQ ******	7, 7.2 (1.1)	5, 5.2 (2.0)
*Social Perception*		
SAT	6, 5.7 (2.4)	4, 5.1 (2.3)
SFRT	6, 5.6 (2.1)	4, 4.1 (1.7)
Biological Motion	5, 4.6 (2.0)	4, 4.0 (1.3)

***** Japanese version of the FEIT, ** Also measures Theory of Mind.

**Table 3 jpm-11-00275-t003:** List and descriptions of the final measures.

Domain/Measure	Description
*Emotion Processing*	
Penn Emotion Recognition Test (ER-40)	Measures the ability to identify emotional state from facial expressions. Participants view 40 still photographs of people’s faces, each expressing a particular emotion (joy, sadness, anger, fear, or no emotion). Participants are then asked to answer, which emotion is expressed in each photograph. Performance is indexed as the number of correct answers. The estimated time required is 3–7 min.
Facial Emotion Selection Test (FEST)	Japanese version of the FEIT. Measures ability to infer emotions from the facial expressions of others. Participants view 21 photographs and answer which emotion (joy, sadness, anger, fear, surprise, disgust, or no emotion) it corresponds to. Performance is indexed as the total number of correct answers. The estimated time required is about 10 min.
Bell Lysaker Emotion Recognition Task-Japanese Version (BLERT-J)	Japanese version of the BLERT. Measures the ability to identify emotional state from facial expression, tone of speech, and body language. Participants view 21 short videos in which an actor portrays different emotional states (happiness, sadness, fear, disgust, surprise, anger, or no emotion) and must answer which emotion was portrayed in each video. Performance is indexed as the number of correct answers. The estimated time required is 7–10 min.
*Theory of Mind*	
Metaphor and Sarcasm Scenario Test (MSST)	Measures ability to understand metaphorical and sarcastic expressions in dialogue. Participants read short passages that provide context for a figurative or sarcastic statement and then choose what they think it means. There are five figurative and five sarcastic statements. The number of correct answers for each type is summed to produce metaphor and sarcasm scores. For each of the sarcasm scenarios, one of the incorrect answers is a “landmine answer” representing the statement’s meaning when taken at face value. The number of times the landmine answer was avoided is tallied as “the landmine avoidance score.” The estimated time required is 5–10 min.
Hinting Task	Measures the ability to detect sarcasm and indirect requests from others’ statements. Participants are read passages of dialogue between two characters in 10 different scenarios. In each conversation, one of the characters tries to indirectly convey a certain intention or request to the other. Participants are asked what the intention or request is. If the answer is incorrect, the participant is provided with additional dialogue that further clarifies the intention. First-time correct answers are awarded two points, and second-time correct answers are awarded one point. Performance is indexed as the total number of points. The estimated time required is about 7 min.
*Attributional Style/Bias*	
Ambiguous Intentions and Hostility Questionnaire (AIHQ)	Assesses hostile social cognitive biases. Participants read passages describing hypothetical, negative scenarios and answer why they think the situation occurred. Participants then rate the degree to which they perceived the action to be intentional, how angry it would make them feel, and how much they would blame the other person on separate Likert scales. Finally, participants answer how they would respond to the situation. Responses to the open-ended questions are coded by independent raters to compute Hostility Bias and Aggression Bias indexes, whereas the Likert ratings are averaged and summed to produce a Blame Score. The estimated time required is 5–10 min.
Intentionality Bias Task (IBT)	Assesses tendency to assign intentionality to the actions of others. Up to 80 short sentences (fewer in some versions) depicting another person’s action (such as “He broke the window”) are presented on a screen. Participants answer whether the behavior is “intentional” or “accidental” within a short time limit. Performance is indexed as the number of questions answered “intentional” to the total number of questions. The estimated time required is about 5 min.
*Social Perception*	
Social Attribution Task-Multiple Choice (SAT-MC)	Assesses implicit social attribution formation. Participants view a 64-s, animated video of anthropomorphized geometric shapes enacting a social drama. The video does not include dialogue. After viewing the video twice, participants answer 19 multiple-choice questions about what happens or how the shapes feel. Performance is indexed as the total number of correct answers. The estimated time required is about 10 min.
Biological Motion (BM) Task	Measures capacity to perceive human body motion at high speed. Participants are presented with images of moving light spots, either moving in coordination to mimic human body movements (Biological Motion) or at random (Scrambled Motion). Participants view multiple images and answer whether each is either Biological Motion or Scrambled Motion. In later parts of the task, random light spots are added/removed in response to correct/incorrect responses to adjust difficulty and determine participants’ level of performance.

## Data Availability

None.

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
