# Peer review of "Evaluation of Social Cognition Measures for Japanese Patients with Schizophrenia Using an Expert Panel and Modified Delphi Method"

_jpm, 2021, doi:10.3390/jpm11040275_

Round 1

Reviewer 1 Report

Dear respected authors, 

This is a very novel piece of research, reported with high-quality communication. 

Some suggestions to make the manuscript stronger: 

Table 2 (and related part of the methodology). You need to report the psychometric goodness of these measures so please add column(s) for reliability, validity, sensitivity, and specificity. Without this information, the readers don't know anything about how good are the measures. This is more important than impact by itself. Also, you need to report if it's self or an expert report; and also the time needed to complete the assessment. You need to discuss these results once you tabulate them. 

Other minor points: 

 in keywords replace QOL with quality of life. 

in Supplementary Materials: The following are available online at www.mdpi.com/xxx/s1, Table S1: 324
List of Items Agreed Upon by the Expert Panel the link does not work. 

Best, 

Reviewer 2 Report

I would like to express profound appreciation for the opportunity to review such outstanding research “Evaluation of social cognition measures for Japanese patients with schizophrenia using an expert panel and modified Delphi method.” This study focused on the cultural difference and identified the measures appropriate for Japanese patients with schizophrenia. Authors assembled the panel of experts reach the set of measures through online voting and conference in modified Delphi method. The background, methods, results, and discussion in the literature are easy to understand and express the importance in this field. Future development of this study is eagerly expected.

The comments and questions are as follows.

1.      Could you explain why authors quoted from “the NIMH Workshop on Social Cognition in Schizophrenia” for the first definition of social cognition as the base of consensus from experts?

2.      In the section of 3.2 of the Results, the target population was specified from “patients with schizophrenia” to “patients with schizophrenia whose symptoms have stabilized following the medication adjustment period in the acute phase and who are undergoing rehabilitation to improve social function”. What was discussed in the course of this process? A final set of measures is possible to be applied to the patients repeatedly to evaluate the rehabilitation effects?

3.      Could you give further explanations of the similarities between the results of this study and the SCOPE study? In related to this, could you explain the necessities of offering the recommendation status in the SCOPE study to the expert panel before rating the appropriateness of the measures?
